# Multi-Scale VMamba: Hierarchy in Hierarchy Visual State Space Model

**Yuheng Shi**
City University of Hong Kong
yuhengshi99@gmail.com

**Minjing Dong**
City University of Hong Kong
minjdong@cityu.edu.hk

**Chang Xu**
University of Sydney
c.xu@sydney.edu.au

## Abstract

Despite the significant achievements of Vision Transformers (ViTs) in various vision tasks, they are constrained by the quadratic complexity. Recently, State Space Models (SSMs) have garnered widespread attention due to their global receptive field and linear complexity with respect to the input length, demonstrating substantial potential across fields including natural language processing and computer vision. To improve the performance of SSMs in vision tasks, a multi-scan strategy is widely adopted, which leads to significant redundancy of SSMs. For a better trade-off between efficiency and performance, we analyze the underlying reasons behind the success of the multi-scan strategy, where long-range dependency plays an important role. Based on the analysis, we introduce Multi-Scale Vision Mamba (MSVMamba) to preserve the superiority of SSMs in vision tasks with limited parameters. It employs a multi-scale 2D scanning technique on both original and downsampled feature maps, which not only benefits long-range dependency learning but also reduces computational costs. Additionally, we integrate a Convolutional Feed-Forward Network (ConvFFN) to address the lack of channel mixing. Our experiments demonstrate that MSVMamba is highly competitive, with the MSVMamba-Tiny model achieving 83.0% top-1 accuracy on ImageNet, 46.9% box mAP, and 42.5% instance mAP with the Mask R-CNN framework, 1x training schedule on COCO, and 47.9% mIoU with single-scale testing on ADE20K. Code is available at https://github.com/YuHengsss/MSVMamba.

## 1 Introduction

In the domain of computer vision, the extraction of features plays a pivotal role in the performance of various tasks, ranging from image classification to more complex applications like detection and segmentation. Traditionally, Convolutional Neural Networks (CNNs) [25, 40, 18, 21, 33] have been the backbone of feature extraction methodologies, prized for their linear scaling complexity and proficiency in capturing local patterns. However, CNNs often fall short in encapsulating global context, a limitation that becomes increasingly apparent in tasks requiring a comprehensive understanding of the entire visual field. In contrast, Vision Transformers (ViTs) [6, 32, 50, 43] have emerged as a compelling alternative, boasting an inherent global receptive field that allows for the direct capture of long-range dependencies within an image. Despite their advantages, ViTs are hampered by their quadratic scaling complexity concerning the input size, which significantly constrains their applicability to downstream tasks such as object detection and segmentation, where efficiency is paramount. Recently, State Space Model (SSM)-based approaches [11, 41, 8] have garnered attention for their ability to combine the best of both worlds: a global receptive field and linear scaling complexity. Notably, Mamba [9] introduces a hardware-aware and input-dependent algorithm that significantly enhances the performance and efficiency of SSMs. Inspired by Mamba's success, a burgeoning body of work has sought to leverage its advantages for vision tasks, pioneering efforts such as ViM [60] and VMamba [30].

38th Conference on Neural Information Processing Systems (NeurIPS 2024).

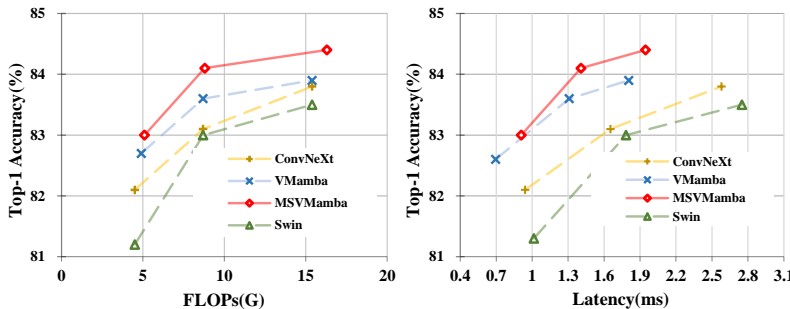

Figure 1: FLOPs and latency comparison on ImageNet. The latency was tested on a RTX 4090 GPU with a batch size of 128 using FP32 precision at an image resolution of 224.

The S6 block, developed by Mamba [9], was originally designed for NLP tasks. To adapt S6 for vision tasks, images are first divided into patches and then flattened into a patch sequence along the scanning path. To accommodate the non-causal nature of image data, the multi-scan strategy is widely adopted for vision tasks, such as ViM [60] which enhances the sequence by summing it in both forward and reverse directions, and VMamba [30] which integrates horizontal and vertical scans. However, unlike NLP models, which can have up to billions of parameters, current vision backbones always take computational costs into consideration, *i.e.*, the trade-off between accuracy and efficiency. This constraint on model size inherently limits the long-range modeling capabilities of SSMs in vision tasks. Taking ViM-Tiny [60] as an example, placing the cls token in the middle of the sequence yields markedly better results than positioning it at the ends. This suggests that central placement compensates for the model's limited ability to integrate distant information, highlighting the difficulties of handling long-range dependencies in parameter-constrained vision models. We refer to this as the long-range forgetting problem. In this work, we analyze how the multi-scan strategy in [60, 30] helps to alleviate this problem. Compared to the single-scan strategy, the multi-scan one allows long-range decay to manifest in various directions within 2D images. However, the increased scanning routes bring multiples of computations, significantly increasing redundancy and limiting efficiency. Thus, we aim to pursue a better trade-off between the performance and efficiency of Mamba in vision tasks.

The most direct and effective method to address the long-range forgetting problem is to shorten the sequence length, which can be achieved by downsampling the feature map. However, placing all scanning routes on a downsampled feature map could result in the loss of fine-grained features and the downstream task performance. Through the visualization of different scans, we show that the decay rates could vary for different scanning routes, which motivates us to develop a hierarchical design of multi-scan. In this work, we propose a Multi-Scale 2D (MS2D) scanning strategy to alleviate the long-range forgetting problem with limited computational costs. Specifically, we divide the scanning directions of SS2D [30] into two groups: one retains the original resolution and is processed by the S6 block, while the others are downsampled, processed by the S6 block, and then upsampled, which not only shortens the sequence length for long-range dependencies learning but also alleviates redundancy. Building on the VMamba with its hierarchical architecture, we incorporate another hierarchical design within the block, creating a hierarchy within a hierarchy. Furthermore, we introduce a Convolutional Feed-Forward Network (ConvFFN) within each block to bolster the model's capability for channel-wise information exchange and local feature capture.

We conduct extensive experiments to validate the effectiveness of MSVmamba across a spectrum of tasks, including image classification, object detection, and semantic segmentation. Detailed comparisons on the ImageNet-1K [3] dataset are illustrated in Fig. 1. MSVMamba achieves a notable improvement over VMamba across different model sizes.

## 2 Related work

### 2.1 Generic Vision Backbone

The evolution of generic vision backbones has significantly shaped the landscape of computer vision, transitioning from CNNs [25, 40, 18, 54, 21, 19, 42, 33] to ViTs [6, 32, 31, 50, 51, 5, 43, 57]. CNNs have been the cornerstone of vision-based models, dominating vision tasks in the early era of deep

learning. The classic CNNs such as AlexNet [25], VGG [40], and ResNet [18], have paved the way for numerous subsequent innovations [54, 21, 19, 42, 20, 33, 4, 49]. These designs have significantly improved performance on a wide range of vision tasks by enhancing the network's ability to capture complex patterns and features from visual data. The Vision Transformer [6], drawing inspiration from the success of transformers [47] in natural language processing, has emerged as a formidable contender to conventional CNNs for vision-related tasks. ViT reimagines image processing by segmenting an image into patches and employing self-attention mechanisms to process these segments. This innovative approach enables the model to discern global dependencies across the entire image, a significant leap forward in understanding complex visual data. However, the ViT architecture demands considerable computational resources and extensive datasets for effective training. Moreover, its performance is intricately tied to the input sequence length, exhibiting a quadratic complexity that can escalate processing costs. In response, subsequent research has focused on developing more efficient training strategies [43, 45, 24], hierarchical network structures [32, 31, 5, 50, 51, 15], refined spatial attention mechanisms [57, 59, 52, 46, 16, 5] and convolution-based design [2, 29, 14, 36] to address these issues.

## 2.2 State Space Models

State Space Models (SSMs) [10, 12, 11, 41, 8] have garnered increasing attention from researchers due to their computational complexity, which grows linearly with the length of the input sequence, and their inherent global awareness properties. To reduce the computational resource consumption of SSMs, S4 [11] introduced a diagonal structure and combined it with a diagonal plus low-rank approach to construct structured SSMs. Subsequently, S5 [41] and H3 [8] further enhanced the efficiency of SSM-based models by introducing parallel scanning techniques and improving hardware utilization. Mamba [9] then introduced the S6 block, incorporating data-dependent parameters to amend the Linear Time Invariant(LTI) characteristics of previous SSM models, demonstrating superior performance over transformers on large-scale datasets. In the realm of vision tasks, S4ND [35] pioneered the application of SSM models in vision tasks by treating visual data as 1D, 2D, and 3D signals. U-Mamba [34] combined CNNs with SSMs for medical image segmentation. ViM [60] and VMamba [30] integrated the S6 block into vision backbone design, employing multiple scanning directions to accommodate the non-casual nature of image data, achieving competitive results against ViTs and CNNs. Motivated by the success of ViM and VMamba, a plethora of Mamba-based works [37, 23, 7, 55, 1, 26, 56, 39] have emerged across various vision tasks, including vision backbone design [37, 23, 55], medical image segmentation [39, 27], and video classification [26], showcasing the potential of SSM-based approaches in advancing the field of computer vision.

## 3 Method

In this section, we first summarize the state space model in Section 3.1. Subsequently, in Section 3.2, we provide an in-depth analysis of the multi-scan strategy in existing vision Mamba models. Following the analysis, Section 3.3 tackles the redundancy and long-range dependency issue by introducing a Multi-Scale 2D (MS2D) scanning strategy. Finally, Section 3.4 details the integration of the Multi-Scale State Space (MS3) block, which incorporates the MS2D technique alongside a ConvFFN. Building upon the MS3 block, various model configurations are developed across different scales, illustrating the adaptability and scalability of our proposed approach.

### 3.1 Preliminaries

**State Space Models.** Classical State Space Models (SSMs) represent a continuous system that maps an input sequence $x(t) \in \mathbb{R}^L$ to a latent space representation $h(t) \in \mathbb{R}^N$ and subsequently predicts an output sequence $y(t) \in \mathbb{R}^L$ based on this representation. Mathematically, an SSM can be described as follows:

$$h'(t) = \mathbf{A}h(t) + \mathbf{B}x(t), \; y(t) = \mathbf{C}h(t), \tag{1}$$

where $\mathbf{A} \in \mathbb{R}^{N \times N}$, $\mathbf{B} \in \mathbb{R}^{N \times 1}$ and $\mathbf{C} \in \mathbb{R}^{1 \times N}$ are learnable parameters.

**Discretization.** To adapt continuous State Space Models (SSMs) for use within deep learning frameworks, it is crucial to implement discretization operations. By incorporating a timescale parameter $\mathbf{\Delta} \in \mathbb{R}$ and employing the widely utilized zero-order hold (ZOH) as the discretization rule,

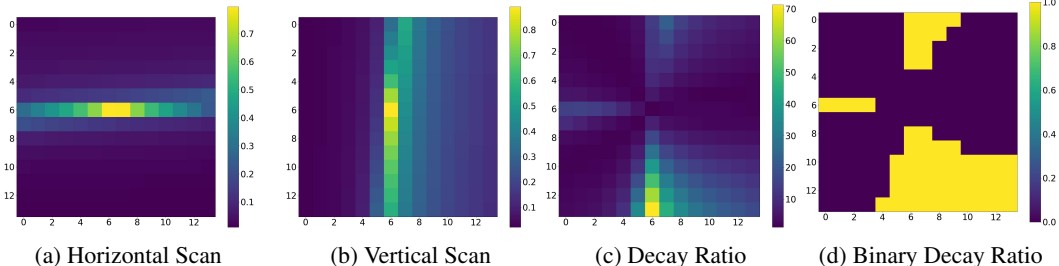

| (a) Horizontal Scan | (b) Vertical Scan | (c) Decay Ratio | (d) Binary Decay Ratio |

Figure 2: Illustration of decay along horizontal, vertical scanning routes and their ratio.

the discretized versions of $\mathbf{A}$ and $\mathbf{B}$ (denoted as $\overline{\mathbf{A}}$ and $\overline{\mathbf{B}}$, respectively) can be derived, with which, Eq. 1 can be reformulated into a discretized manner as:

$$
\begin{aligned}
h(t) &= \overline{\mathbf{A}}h(t-1) + \overline{\mathbf{B}}x(t), \ y(t) = \mathbf{C}h(t), \\
\text{where } \overline{\mathbf{A}} &= e^{\mathbf{\Delta A}}, \ \overline{\mathbf{B}} = (\mathbf{\Delta A})^{-1}(e^{\mathbf{\Delta A}} - \mathbf{I})\mathbf{\Delta B} \approx \mathbf{\Delta B},
\end{aligned}
\tag{2}
$$

where $\mathbf{I}$ denotes the identity matrix. Afterward, the process of Eq. 2 could be implemented in a global convolution manner as:

$$
y = x \odot \overline{\mathbf{K}}, \ \overline{\mathbf{K}} = \left( \mathbf{C}\overline{\mathbf{B}}, \mathbf{C}\overline{\mathbf{A}}\overline{\mathbf{B}}, \dots, \mathbf{C}\overline{\mathbf{A}}^{L-1}\overline{\mathbf{B}} \right),
\tag{3}
$$

where $\overline{\mathbf{K}} \in \mathbb{R}^L$ is the convolution kernel.

**Selective State Space Models.** The Selective State Space (S6) mechanism, introduced by Mamba [9], renders the parameters $\overline{\mathbf{B}}, \overline{\mathbf{C}}$, and $\mathbf{\Delta}$ input-dependent, thereby enhancing the performance of SSM-based models. After making $\overline{\mathbf{B}}, \overline{\mathbf{C}}$, and $\mathbf{\Delta}$ input-dependent, the global convolution kernel in Eq. 3 could be rewritten as:

$$
\overline{\mathbf{K}} = \left( \mathbf{C}_L\overline{\mathbf{B}}_L, \mathbf{C}_L\overline{\mathbf{A}}_{L-1}\overline{\mathbf{B}}_{L-1}, \dots, \mathbf{C}_L \prod_{i=1}^{L-1} \overline{\mathbf{A}}_i\overline{\mathbf{B}}_1 \right).
\tag{4}
$$

### 3.2 Analysis of Multi-Scan Strategy

When processing image data using the S6 block, the 2D feature map $\mathbf{Z} \in \mathbb{R}^{H \times W \times D}$ is flattened into a 1D sequence of image tokens, denoted as $\mathbf{X} \in \mathbb{R}^{L \times D}$. According to Eq. 4 and Eq. 2, the contribution of the $m_{th}$ token to the construction of the $n_{th}$ token ($m < n$) in S6 can be expressed as:

$$
\mathbf{C}_n \prod_{i=m}^{n} \overline{\mathbf{A}}_i\overline{\mathbf{B}}_m = \mathbf{C}_n\overline{\mathbf{A}}_{(m \to n)}\overline{\mathbf{B}}_m, \ \text{where } \overline{\mathbf{A}}_{(m \to n)} = e^{\sum_{i=m}^{n} \mathbf{\Delta}_i \mathbf{A}}.
\tag{5}
$$

Typically, the learned $\mathbf{\Delta}_i\mathbf{A}$ is negative, which biases the model towards prioritizing recent tokens' information. Consequently, as the sequence length increases, the exponential term $e^{\sum_{i=m}^{n} \mathbf{\Delta}_i \mathbf{A}}$ in Eq. 5 decays significantly, resulting in minimal contributions from distant tokens. We refer to it as the long-range forgetting issue, which has also been observed in recent studies applying S6 to vision tasks [13]. Although this problem can be mitigated by increasing the number of parameters and the depth of the model, such adjustments introduce additional computational costs. Furthermore, the causal property of the S6 block ensures that information can only propagate in a unidirectional manner between tokens, preventing earlier tokens from accessing information from subsequent tokens.

The inherent non-causal nature of images renders the direct application of the S6 block to vision-related tasks less than optimal, as identified by ViM [60]. To mitigate this limitation, ViM [60] and VMamba [30] have introduced methodologies that entail scanning image features across various directions and then integrating these features. Generally, the updated token along one of the scanning routes, denoted as $Scan(\mathbf{Z}_{(p,q)})$, where $(p,q)$ indicates the coordinate, could be obtained by:

$$
Scan(\mathbf{Z}_{(p,q)}) = \mathbf{C}_\alpha \sum_{i=1}^{\alpha} \overline{\mathbf{A}}_{(i \to \alpha)}\overline{\mathbf{B}}_i\sigma(\mathbf{Z})_i.
\tag{6}
$$

In Eq. 6, $\sigma$ represents the transformation that converts a 2D feature map into a 1D sequence, and $\alpha$ denotes the corresponding index of $\mathbf{Z}(p,q)$ in the transformed 1D sequence. Afterward, results from

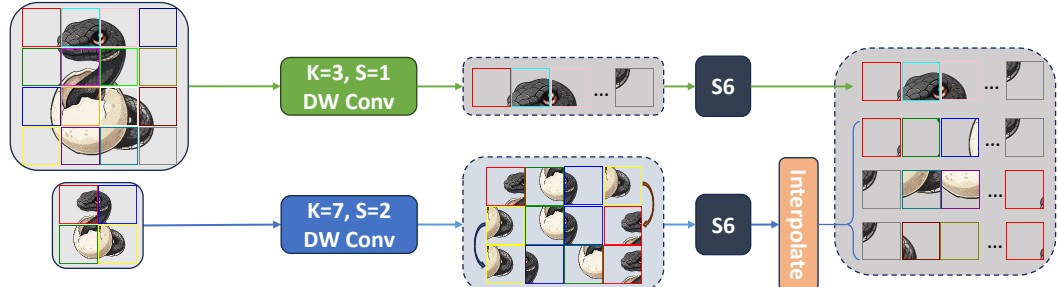

Figure 3: Illustration of the Multi-Scale 2D-Selective-Scan on an image

multi-scan routes are added together to produce enhanced feature $\mathbf{Z}'(p, q)$, which can be denoted as:

$$\mathbf{Z}'_{(p,q)} = \sum_k Scan_k(\mathbf{Z}_{(p,q)}) = \sum_k \mathbf{C}_{\alpha_k} \sum_{i=1}^{\alpha_k} \overline{\mathbf{A}}_{(i \to \alpha_k)} \overline{\mathbf{B}}_i \sigma_k(\mathbf{Z})_i. \tag{7}$$

This multi-scan strategy allows tokens to access information from each other. In ViM [60], two distinct scanning routes correspond to two different transformations in Eq. 6, specifically, the flatten and the flatten with flip transformations. Similarly, VMamba [30] extends the basic bidirectional scanning by incorporating both horizontal and vertical scanning directions, yielding four distinct scanning routes. Besides, the multi-scan strategy also alleviates the long-range forgetting problem by minimizing the effective distance between tokens. For tokens at coordinates $(p_1, q_1)$ and $(p_2, q_2)$, the strategy employs multiple scanning routes, each potentially altering their relative positions. The minimum distance across these routes is given by $\min_k d_k((p_1, q_1) \to (p_2, q_2))$, where $d_k$ represents the distance between the tokens in the sequence generated by the $k$-th scan. By reducing this distance, the multi-scan strategy reduces the decay of influence between distant tokens, thereby enhancing the model's ability to maintain and utilize long-range information.

To more intuitively demonstrate the relationship between the multi-scan strategy and long-range decay, we visualize the exponential term $e^{\sum_{i=1}^{\alpha} \Delta_i \mathbf{A}}$ along the horizontal and vertical scanning directions in VMamba-Tiny with respect to the central token in Fig. 2a and Fig. 2b. Specifically, we randomly select 50 images from the ImageNet [3] validation set and compute the average decay along scanning routes at the last layer of the final stage across these images and feature dimensions. We use a higher input resolution to enhance the quality of the visualization.

According to these observations, the success of the multi-scan strategy in VMamba can be attributed to its mitigation of the non-causal properties of image data and alleviation of the long-range forgetting problem. However, as the number of scanning routes increases, the computational cost also rises linearly, introducing computational redundancy. In Fig. 2c, we illustrate the maximum ratio of Fig. 2a to Fig. 2b and vice versa. While in Fig. 2d, we present a binarized version of Fig. 2c, applying a threshold of 10, which covers more than 40% of the entire figure. This phenomenon indicates that the varying decay rates across different scanning routes lead to certain routes dominating the decay dynamics, which can also be attributed to the long-range forgetting problem. The existence of dominant scanning routes implies that some scans contribute significantly more to information retention than others, leading to computational redundancy in the multi-scan strategy.

## 3.3 Multi-Scale 2D Scanning

As discussed in the last subsection, the contribution of tokens decays with increasing scanning distance. The most effective and direct way to alleviate the long-range forgetting problem is to reduce the number of tokens. Simultaneously, since the computational complexity of the S6 block is linearly dependent on the number of tokens, reducing the token count also enhances efficiency. Thus, an alternative approach to address the aforementioned issue is to apply the multi-scan strategy on a downsampled feature map. However, setting all scans on a downsampled feature map will ignore fine-grained features and result in unavoidable information loss. Thus, scanning along the full-resolution feature map is also essential.

Motivated by these considerations, we introduce a simple yet effective Multi-Scale 2D scanning(MS2D) strategy, as depicted in Fig. 3. Our approach commences with the generation of

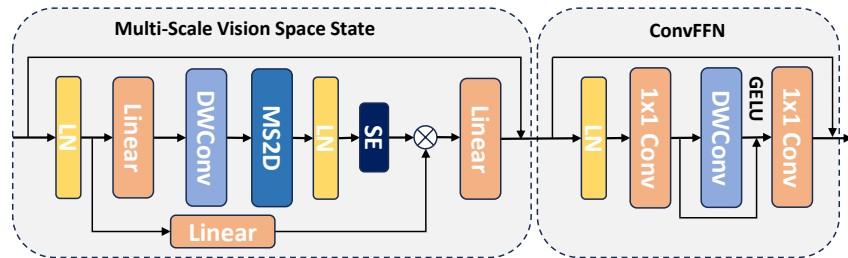

Figure 5: Detailed architecture of Multi-Scale State Space (MS3) block, consisting of a Multi-Scale Vision Space State (MSVSS) block and a Convolutional Feed-Forward Network (ConvFFN) block.

hierarchical feature maps at varying scales, achieved through the application of Depthwise Convolution (DWConv) with distinct stride values. These multi-scale feature maps are then processed through four distinct scanning routes within VMamba. Specifically, we uitlize DWConvs with strides of 1 and $s$ to obtain feature map $\mathbf{Z}_1 \in \mathbb{R}^{H \times W \times D}$ and $\mathbf{Z}_2 \in \mathbb{R}^{\frac{H}{s} \times \frac{W}{s} \times D}$, respectively. Afterwards, $\mathbf{Z}_1$ and $\mathbf{Z}_2$ are processed by two S6 blocks as:

$$\mathbf{Y}_1 = S6(\sigma_1(\mathbf{Z}_1)), \tag{8}$$

$$[\mathbf{Y}_2, \mathbf{Y}_3, \mathbf{Y}_4] = S6([\sigma_2(\mathbf{Z}_2), \sigma_3(\mathbf{Z}_2), \sigma_4(\mathbf{Z}_2)]), \tag{9}$$

where $\sigma$ is transformation that convert 2D feature maps into 1D sequences used in SS2D, and $\mathbf{Y}$ is the processed sequence. These processed sequences are converted back into 2D feature maps, and the downsampled feature maps are interpolated for merging:

$$\mathbf{Z}'_i = \gamma_i(\mathbf{Y}_i), i \in \{1, 2, 3, 4\}, \tag{10}$$

$$\mathbf{Z}' = \mathbf{Z}'_1 + \text{Interpolate}(\sum(\mathbf{Z}'_j)), j \in \{2, 3, 4\}, \tag{11}$$

where $\gamma$ is the inverse transformation of $\sigma$ and $\mathbf{Z}'$ is the feature map enhanced by MS2D. The downsampling operation reduces the sequence length by a factor of $s^2$, which also shortens the distance between tokens in Eq. 5 by a factor of $s^2$, thereby alleviating the long-range forgetting problem. As the computational complexity of a single S6 block is $O(9LDN)$ [9], where $N$ denotes the SSM dimension, replacing SS2D with MS2D reduces the total sequence length across four scans from $4L$ to $(1+3/s^2)L$, thereby improves the efficiency. Practically, the downsampling rate is set to 2. It is worth noting that sequences from $\mathbf{Z}_2$ are processed by the same S6 block. This approach

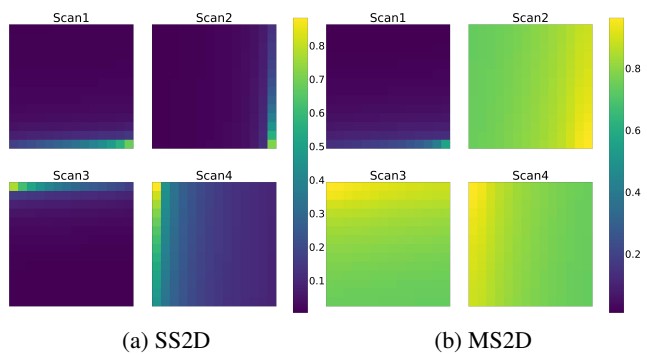

(a) SS2D      (b) MS2D

Figure 4: Illustration of the decay with different scanning routes in SS2D and MS2D.

maintains the same accuracy as using multiple S6 blocks for different scanning routes while effectively reducing the number of parameters.

To better illustrate the alleviation of the long-range forgetting problem, we also provide empirical evidence, as shown in Fig. 4. We compare the decay along scanning routes in the SS2D of VMamba and our MS2D, focusing on the last token with the same configuration as Fig. 2. The decay maps in downsampled features are interpolated back. As observed, the decay rate along scanning routes in downsampled maps is significantly alleviated, enhancing the capability to capture global information.

### 3.4 Overall Model Architecture

In this study, we extend the capabilities of the VMamba framework by substituting its VSS block with our Multi-Scale State Space (MS3) block. The architectural framework of the MS3 block is delineated in Fig. 5, comprising a Multi-Scale Vision Space State(MSVSS) component and a Convolutional Feed-Forward Network (ConvFFN). The MSVSS component is devised by adapting the vision

state space framework in VMamba, substituting the SS2D with an MS2D to further introduce a hierarchy design in a single layer. Additionally, a Squeeze-Excitation (SE) [20] block is integrated subsequent to the multi-scale 2D scanning, as informed by related literature [23, 37]. Diverging from the conventional focus on token mixing in prior vision Mamba architectures [60, 30, 23], our design introduces a channel mixer to augment the flow of information across different channels, aligning with the structural paradigms of typical vision transformers. In concordance with preceding studies [52, 48, 14, 51], the ConvFFN which consists of a depth-wise convolution and two fully connected layers is employed as the channel mixer. Upon the amalgamation of MSVSS and ConvFFN within the MS3 block, meticulous adjustments are made to the number of blocks to ensure a comparable computational budget, facilitating a fair comparison.

To empirically validate the efficacy of our proposed modifications, we introduce model variants across different scales. These variants, namely Nano, Micro, Tiny, Small and Base, are characterized by their parameter counts of 7M, 12M, 32M, 50M and 88M respectively. In terms of computational expenditure, these models require 0.9, 1.5, 5.1, 8.8 and 15.5 GFLOPs correspondingly, demonstrating a scalable approach to model design that accommodates varying computational constraints. For models above tiny size, the multiplicative branch of the MSVSS block is removed as informed by the VMambav3 [30] for better performance. Detailed architectures are shown in Appendix A.

## 4 Experimental Validation

### 4.1 ImageNet Classification

**Settings.** Our models are trained and tested on the ImageNet-1K dataset [3]. In alignment with previous works [32, 30, 23], all models undergo training for 300 epochs, with the initial 20 epochs dedicated to warming up. The training utilizes a batch size of 1024 across 8 GPUs. We employ the AdamW optimizer, setting the betas to (0.9, 0.999) and momentum to 0.9. The learning rate is managed through a cosine decay scheduler, starting from an initial rate of 0.001, coupled with a weight decay of 0.05. Additionally, we leverage the exponential moving average (EMA) and implement label smoothing with a factor of 0.1 to enhance model performance and generalization. During testing, images are center cropped with the size of $224 \times 224$. When dealing with the routes for multi-scale scanning, we select top-left to the bottom-right for dealing with the full-resolution feature map while the other three scans are responsible for scanning the downsampled feature map.

**Results.** Tab. 1 showcases our MSV-Mamba models against established CNNs, ViTs, and SSM-based models on ImageNet-1K. MSVMamba-T, with 32M parameters and 5.1G FLOPs, achieves 83.0% top-1 accuracy, outperforming similar-cost SSM-based LocalVMamba-T. The MSVMamba-B model

Table 1: Accuracy comparison across various models on ImageNet-1K.

| Method | #param. | FLOPs | Top-1 Acc(%). |
|---|---|---|---|
| RegNetY-800M [38] | 6M | 0.8G | 76.3 |
| RegNetY-1.6G [38] | 11M | 1.6G | 78.0 |
| RegNetY-4G [38] | 21M | 4.0G | 80.0 |
| DeiT-S [43] | 22M | 4.6G | 79.8 |
| DeiT-B [43] | 86M | 17.5G | 81.8 |
| Swin-T [32] | 29M | 4.5G | 81.3 |
| Swin-S [32] | 50M | 8.7G | 83.0 |
| Swin-B [32] | 88M | 15.4G | 83.5 |
| ViM-T [60] | 7M | 1.5G | 76.1 |
| ViM-S [60] | 26M | 5.1G | 80.5 |
| VMambav3-T [30] | 30M | 4.9G | 82.6 |
| VMambav3-S [30] | 50M | 8.7G | 83.6 |
| VMambav3-B [30] | 89M | 15.4G | 83.9 |
| LocalVMamba-T [23] | 26M | 5.7G | 82.7 |
| LocalVMamba-S [23] | 50M | 11.4G | 83.7 |
| MSVMamba-N | 7M | 0.9G | 77.3 |
| MSVMamba-M | 12M | 1.5G | 79.8 |
| MSVMamba-T | 32M | 5.1G | 83.0 |
| MSVMamba-S | 50M | 8.8G | 84.1 |
| MSVMamba-B | 91M | 16.3G | 84.4 |

attains 84.4% accuracy with 91M parameters and 16.3G FLOPs, exceeding VMambav3-B by 0.5%. These results highlight MSVMamba's efficiency and scalability, offering a robust option for high-accuracy, resource-efficient model design.

### 4.2 Object Detection

**Setup.** We evaluate our MSVMamba on the MSCOCO [28] dataset using the Mask R-CNN [17] framework for object detection and instance segmentation tasks. Following previous works [32, 30],

Table 2: Object detection and instance segmentation with Mask R-CNN on COCO. The FLOPs are computed for an input size of $1280 \times 800$. Multi-scale training is exclusively implemented in the 3× schedule. All backbones are pre-trained on the ImageNet-1K dataset.

| Backbone | Mask R-CNN 1x Schedule | | | | | | Mask R-CNN 3x Schedule | | | | | | #param. | FLOPs |
|---|---|---|---|---|---|---|---|---|---|---|---|---|---|---|
| | $AP^b$ | $AP^b_{50}$ | $AP^b_{75}$ | $AP^m$ | $AP^m_{50}$ | $AP^m_{75}$ | $AP^b$ | $AP^b_{50}$ | $AP^b_{75}$ | $AP^m$ | $AP^m_{50}$ | $AP^m_{75}$ | | |
| PVT-T [50] | 36.7 | 59.2 | 39.3 | 35.1 | 56.7 | 37.3 | 39.8 | 62.2 | 43.0 | 37.4 | 59.3 | 39.9 | 33M | 208G |
| LightViT-T [22] | 37.8 | 60.7 | 40.4 | 35.9 | 57.8 | 38.0 | 41.5 | 64.4 | 45.1 | 38.4 | 61.2 | 40.8 | 28M | 187G |
| EffVMamba-S [37] | 39.3 | 61.8 | 42.8 | 36.7 | 58.9 | 39.2 | 41.6 | 63.9 | 45.6 | 38.2 | 60.8 | 40.7 | 31M | 197G |
| MSVMamba-M | 43.8 | 65.8 | 47.7 | 39.9 | 62.9 | 42.9 | 46.3 | 68.1 | 50.8 | 41.8 | 65.1 | 44.9 | 32M | 201G |
| Swin-T [32] | 42.7 | 65.2 | 46.8 | 39.3 | 62.2 | 42.2 | 46.0 | 68.1 | 50.3 | 41.6 | 65.1 | 44.9 | 48M | 267G |
| ConvNeXt-T [33] | 44.2 | 66.6 | 48.3 | 40.1 | 63.3 | 42.8 | 46.2 | 67.9 | 50.8 | 41.7 | 65.0 | 44.9 | 48M | 262G |
| VMambav3-T [30] | 47.3 | 69.3 | 52.0 | 42.7 | 66.4 | 45.9 | 48.8 | 70.4 | 53.5 | 43.7 | 67.4 | 47.0 | 50M | 271G |
| LocalVMamba-T [23] | 46.7 | 68.7 | 50.8 | 42.2 | 65.7 | 45.5 | 48.7 | 70.1 | 53.0 | 43.4 | 67.0 | 46.4 | 45M | 291G |
| MSVMamba-T | 46.9 | 68.7 | 51.4 | 42.5 | 66.2 | 45.8 | 48.7 | 69.8 | 53.3 | 43.4 | 67.2 | 46.8 | 52M | 275G |
| Swin-S [32] | 44.8 | 66.6 | 48.9 | 40.9 | 63.2 | 44.2 | 48.2 | 69.8 | 52.8 | 43.2 | 67.0 | 46.1 | 69M | 354G |
| ConvNeXt-S [33] | 45.4 | 67.9 | 50.0 | 41.8 | 65.2 | 45.1 | 47.9 | 70.0 | 52.7 | 42.9 | 66.9 | 46.2 | 70M | 348G |
| VMambav3-S [30] | 48.7 | 70.0 | 53.4 | 43.7 | 67.3 | 47.0 | 49.9 | 70.9 | 54.7 | 44.2 | 68.2 | 47.7 | 70M | 349G |
| MSVMamba-S | 48.1 | 70.1 | 52.8 | 43.2 | 67.3 | 46.5 | 49.7 | 70.9 | 54.3 | 44.2 | 68.0 | 47.9 | 70M | 349G |

we utilize backbones pretrained on ImageNet-1K for initialization. We employ standard training strategies of $1\times$ (12 epochs) and $3\times$ (36 epochs) with Multi-Scale (MS) training for a fair comparison.

**Results.** Tab. 2 presents a performance comparison of our method against CNNs, ViTs, and SSM-based models. Our model achieve competitive results across various variants and training settings. Specifically, MSVMamba-T outperforms Swin-T by +4.2 box AP and +3.3 mask AP under the $1\times$ schedule and also shows improvements in both box AP and mask AP under the $3\times$ schedule.

## 4.3 Semantic Segmentation

**Setup.** Consistent with the methodologies used in Swin [32] and VMamba [30], we utilize the UperHead [53] framework atop an ImageNet pre-trained MSVMamba backbone. The training process is conducted over 160K iterations with a batch size of 16. We employ the AdamW optimizer with a learning rate set at $6 \times 10^{-5}$. Our experiments are primarily conducted using a default input resolution of $512 \times 512$. Additionally, we also incorporate Multi-Scale (MS) testing to assess performance variations.

**Results.** We present the detailed results of our model and other competitors in Tab. 3, which includes both single-scale and multi-scale testing. Our MSVMamba consistently outperforms the Swin and ConNeXt models in the tiny variant by margins of +2.2 and +1.6 mIoU, respectively.

Table 3: We present the results of semantic segmentation on the ADE20K dataset [58] using the UperNet framework [53]. The computational complexity, measured in FLOPs, is calculated for input dimensions of $512 \times 2048$. The abbreviations "SS" and "MS" refer to single-scale and multi-scale testing, respectively.

| Method | mIoU SS | mIoU MS | #param. | FLOPs |
|---|---|---|---|---|
| ResNet-50 [18] | 42.1 | 42.8 | 67M | 953G |
| DeiT-S+MLN [44] | 43.8 | 45.1 | 58M | 1217G |
| Swin-T [32] | 44.4 | 45.8 | 60M | 945G |
| ConvNeXt-T [33] | 46.0 | 46.7 | 60M | 939G |
| VMambav3-T [30] | 47.9 | 48.8 | 62M | 949G |
| MSVMamba-M | 45.1 | 45.4 | 42M | 875G |
| MSVMamba-T | 47.9 | 48.5 | 63M | 953G |

## 4.4 Ablation Study

To validate the effectiveness of the proposed modules, we conducted a comprehensive ablation study. Specifically, we scaled the VMamba-Tiny model by setting its embedding dimension $d$ to 48, the state space dimension $N$ to 8, and the number of blocks in the four different stages to $[1, 2, 4, 2]$. The scaled model, referred to as VMamba-Nano, has parameters and computational costs of 4.4M and 0.87GFLOPs, respectively. This model and the standard tiny-sized models serve as the baseline for our ablation experiments. Models in ablation study are conducted over a training schedule of 100 epochs on ImageNet-1K to reduce training time. Besides, the $AP_b$ and $AP_m$ under Mask R-CNN with 1x schedule on COCO dataset are also reported for nano-size models.

Table 4: Evolutionary trajectory from VMamba to MSVMamba on nano-sized model.

| Model | MS2D | SE | ConvFFN | $N=1$ | #param. | FLOPs | Top-1 Acc(%). | $AP_b$ | $AP_m$ |
|---|---|---|---|---|---|---|---|---|---|
| VMamba-Nano | | | | | 4.4M | 0.87G | 69.6 | 38.1 | 35.6 |
| MSVMamba-Nano | ✓ | | | | 4.8M | 0.89G | $71.9_{\uparrow 2.3}$ | $39.1_{\uparrow 1.0}$ | $36.3_{\uparrow 0.7}$ |
| | ✓ | ✓ | | | 5.3M | 0.89G | $72.4_{\uparrow 2.8}$ | $39.5_{\uparrow 1.4}$ | $36.5_{\uparrow 0.9}$ |
| | ✓ | ✓ | ✓ | | 6.6M | 0.94G | $74.4_{\uparrow 4.8}$ | $41.0_{\uparrow 2.9}$ | $37.8_{\uparrow 2.2}$ |
| | ✓ | ✓ | ✓ | ✓ | 6.9M | 0.88G | $75.1_{\uparrow 5.5}$ | $41.4_{\uparrow 3.3}$ | $37.9_{\uparrow 2.3}$ |

Table 5: Performance ablation on tiny-size models. FPS and Memory are tested on a 4090 GPU with a batch size of 128 and FP32 precision. The symbol † indicates model inherit optimization used in VMambav3 version.

| Model | MS2D | SE | ConvFFN | $N=1$ | #param. | FLOPs | Top-1 Acc(%). | FPS | Memory (MB) |
|---|---|---|---|---|---|---|---|---|---|
| VMambav1-Tiny | | | | | 23M | 5.6G | 80.3 | 603 | 6639 |
| MSVMamba-Tiny | ✓ | | | | 24M | 4.8G | $80.9_{\uparrow 0.6}$ | 866 | 4780 |
| | ✓ | ✓ | ✓ | ✓ | 33M | 4.6G | $81.4_{\uparrow 1.1}$ | 1092 | 4533 |
| MSVMamba-Tiny† | ✓ | ✓ | ✓ | ✓ | 32M | 5.1G | $81.7_{\uparrow 1.4}$ | 1097 | 2413 |

**On Multi-Scale 2D Scan.** For the nano-size variant, replacing SS2D with our MS2D in the VMamba framework resulted in an increase in accuracy on ImageNet-1K from 69.6% to 71.9%. Additionally, the $AP_b$ and $AP_m$ metrics improved from 38.1 and 35.6 to 39.1 and 36.3, respectively, as shown in Tab. 4. Furthermore, we conducted an ablation on the number of scans in the multi-scale scan, considering both full-resolution and half-resolution branches on nano-size models. The results are shown in Tab. 6. Placing all scans in the half-resolution branch led to a significant loss of fine-grained features, resulting in a substantial decrease in model accuracy. Positioning two or three scans in the full resolution branch, compared to just one, resulted in accuracy improvements of 0.1% and 0.6%, respectively, but introduced an additional computational cost of approximately 12% and 25%. Allocating four scans to the full resolution branch, effectively reverting to the SS2D method, increased the computational cost by 34% while only improving accuracy by 0.4%.

For an optimal trade-off between computational cost and accuracy, we select one scan in the full-resolution branch as the default setting. Building upon the MS2D foundation, we introduce an SE block following EfficientV-Mamba [37], which further enhanced accuracy by 0.5% with minimal additional computational cost.

Experiments related to tiny-size model are reported in Tab. 5. Our findings indicate that the proposed MS2D module contributes to an improvement of 0.6% in Top-1 accuracy for the tiny-size model while other components of our model collectively contribute an additional 0.5% increase in accuracy. The MS2D module not only enhances

Table 6: Ablation study on MS2D.

| Full | Half | #param | FLOPs | Top-1 Acc(%). |
|---|---|---|---|---|
| 0 | 4 | 5.1M | 0.74G | 63.1 |
| 1 | 3 | 4.8M | 0.89G | 71.9 |
| 2 | 2 | 5.0M | 1.00G | 72.0 |
| 3 | 1 | 5.3M | 1.11G | 72.5 |
| 4 | 0 | 5.1M | 1.19G | 72.3 |

performance but also contributes to further speed gains and reductions in memory usage. Since MS2D is orthogonal to the updates in VMambav3, they can be combined for further enhancements. In the last row of Tab. 5, we present the results after adopting optimization used in VMambav3, which include reducing the $ssm\ ratio$ and eliminating the entire multiplicative branch.

**On ConvFFN.** Upon replacing SS2D with MS2D and incorporating a SE block, we constructed a model that utilizes ConvFFN as a channel mixer. When only using SSM, the model exhibited insufficient information exchange between channels. The integration of ConvFFN as a channel mixer significantly enhanced the model's capability for inter-channel information interaction. As indicated in Tab. 4, the addition of ConvFFN resulted in an additional accuracy improvement of 2.0%. Besides, we set the state space dimension $N=1$ and stacked one more block to further enhance the capability of capturing long-range information while maintaining a roughly constant computational cost. This operation resulted in an additional accuracy improvement of 0.7%, as shown in Tab. 4. To maintain a

roughly equivalent computational cost, we adjusted the number of blocks within the model. When integrating the ConvFFN and setting the state space dimension $N = 1$, we meticulously calibrate the quantity of blocks to maintain a nearly constant computational cost, measured in GFLOPs.

## 5 Limitations

The design of multi-scale VMamba aims to tackle the long-range forgetting problem of Mamba models with limited parameters on vision tasks. Although the proposed model has proven to be effective, its scalability remains to be explored since this issue can also be alleviated by increasing model sizes.

## 6 Conclusion

In this paper, we introduced Multi-Scale VMamba (MSVMamba), an SSM-based vision backbone that leverages the advantages of linear complexity and global receptive field. We developed the Multi-Scale 2D (MS2D) scanning technique to minimize computational redundancy and alleviate the long-range forgetting problem in parameter-limited vision models. Additionally, we incorporated the Convolutional Feed-Forward Network (ConvFFN) to enhance the exchange of information between channels, thereby significantly improving the performance of our model. Our experiments demonstrate that MSVMamba consistently outperforms popular models from various architectures, including ConvNeXt, Swin Transformer, and VMamba, in image classification and downstream tasks.

## 7 Acknowledgements

This work was supported in part by the Australian Research Council under Projects DP240101848 and FT230100549, and by the Start-up Grant (No. 9610680) of the City University of Hong Kong.

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

# A    Network Architecture

Table 7: **Specifications of MSVMamba varints.**

| Model | Blocks | Channels | $ssm\ ratio$ | $FFN\ ratio$ | **#param.(M)** | **GFLOPs** |
|---|---|---|---|---|---|---|
| **N**ano | [1, 2,  5, 2] | [ 48,  96, 192,  384] | 2 | 2 | 7 | 0.9 |
| **M**icro | [1, 2,  5, 2] | [ 64, 128, 256,  512] | 2 | 2 | 12 | 1.5 |
| **T**iny | [2, 2,  9, 2] | [ 96, 192, 384,  768] | 1 | 4 | 32 | 5.1 |
| **S**mall | [2, 3, 20, 2] | [ 96, 192, 384,  768] | 1 | 4 | 50 | 8.8 |
| **B**ase | [2, 4, 18, 2] | [128, 256, 512, 1024] | 1 | 4 | 91 | 16.3 |

In Tab. 7, we present the detailed architecture of our model variants, including the Nano, Micro, Tiny, Small and Base versions, each with varying channels, block numbers $ssm\ ratio$ and $FFN\ ratio$.

# B    More Ablations

To further demonstrate the effectiveness of our MS2D, we add more baselines that involve only Scan1 (Uni-directional Scan) and a combination of Scan1 and Scan3 (Bi-directional Scan). These results are presented in Tab. 8. Concretely, our MS2D further outperform Uni-directional Scan and Bi-directional Scan baselines by 3.0% and 2.4% top-1 acc respectively.

In Tab. 9, we explore the impact of full-resolution scanning directions. The results indicate that while different scans yield similar accuracy, Scan1 was selected for its marginally superior performance consistency.

Table 8: Ablation with more baselines. The CrossScan is utilized in VMamba, while the Uni-Scan and Bi-Scan denote Uni-directional Scan and Bi-directional Scan respectively.

| Setting | #param.(M) | GFLOPs | Accuracy (%) |
|---|---|---|---|
| Uni-Scan | 4.4 | 0.87 | 68.9 |
| Bi-Scan | 4.4 | 0.87 | 69.5 |
| CrossScan | 4.4 | 0.87 | 69.6 |
| **MS2D** | 4.8 | 0.89 | **71.9** |

Table 9: Ablation for full-resolution branch.

| Full | Scan1 | Scan2 | Scan3 | Scan4 |
|---|---|---|---|---|
| Top-1 Acc(%) | 71.9 | 71.8 | 71.8 | 71.9 |

# C    Efficiency Comparison

We report detailed throughput comparison of Swin [32], ConvNeXt [33], and VMamba [30] in Tab. 10. All models are tested on a RTX 4090 GPU with a batch size of 128 and FP32 precision at an image resolution of 224.

# D    Qualitative Analysis

We provide visualizations in Fig. 6 comparing the proposed MS2D and SS2D configurations in VMamba. These visualizations are generated by converting the S6 layer into an attention format, as demonstrated by VMamba [30]. The results clearly show that the full-resolution in MS2D scan captures more detailed features, whereas the scans at half resolution primarily focus on broader architectural details, compared to SS2D. The proposed hierarchical scanning pattern facilitates the current layer's ability to discern and amalgamate features across various levels of abstraction.

Table 10: Efficiency comparison with our baseline VMamba [30] and widely-used Swin Transformer [32] and ConvNeXt [33].

| Model | Top-1 Acc(%) | #Params | FLOPs (G) | Thru. (imgs/sec) | Memory (MB) |
|---|---|---|---|---|---|
| Swin-T [32] | 81.3 | 28 M | 4.5 | 986 | 2402 |
| ConvNeXt-T [33] | 82.1 | 29 M | 4.5 | 1062 | 1670 |
| VMambav1-T [30] | 82.2 | 23 M | 5.6 | 603 | 6639 |
| VMambav3-T [30] | 82.6 | 30 M | 4.9 | 1456 | 3204 |
| MSVMamba-T | 83.0 | 32 M | 5.1 | 1097 | 2413 |
| Swin-S [32] | 83.0 | 50 M | 8.7 | 561 | 2596 |
| ConvNeXt-S [33] | 83.1 | 50 M | 8.7 | 605 | 1753 |
| VMambav1-S [30] | 83.5 | 44 M | 11.2 | 425 | 6882 |
| VMambav3-S [30] | 83.6 | 50 M | 8.7 | 764 | 5780 |
| MSVMamba-S | 84.1 | 50 M | 8.8 | 708 | 2545 |
| Swin-B [32] | 83.5 | 88 M | 15.5 | 363 | 3362 |
| ConvNeXt-B [33] | 83.8 | 89 M | 15.4 | 387 | 2380 |
| VMambav1-B [30] | 83.7 | 76 M | 18.0 | 314 | 8853 |
| VMambav3-B [30] | 83.9 | 89 M | 15.4 | 555 | 7826 |
| MSVMamba-B | 84.4 | 91 M | 16.3 | 514 | 3699 |

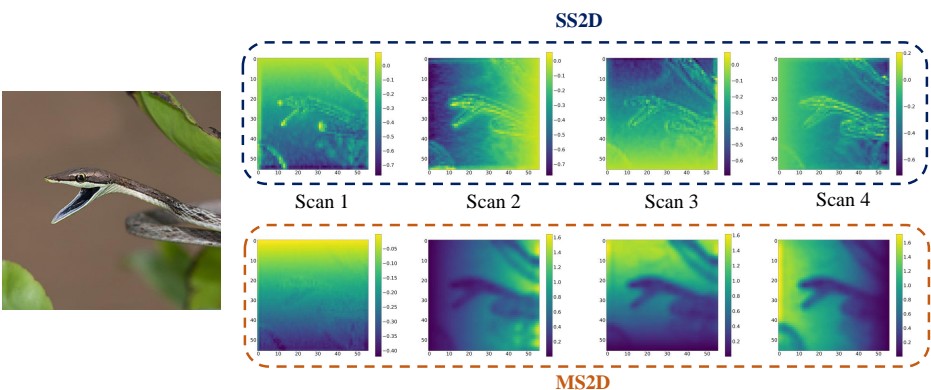

Figure 6: Attention maps from four distinct scanning directions, generated by SS2D and our MS2D in the last layer of the second stage. In the second row, full-resolution scan (first scan) captures fine-grained features, whereas scans at half resolution capture coarse-grained features. Maps are rendered at a higher resolution to enhance visualization quality.

