# OpenReview forum: "Multi-Scale VMamba: Hierarchy in Hierarchy Visual State Space Model"
_NeurIPS.cc/2024/Conference — NeurIPS 2024 poster_

### Official Review · Reviewer_HsBE · 2024-07-10

**Soundness:** 3
**Presentation:** 3
**Contribution:** 3
**Rating:** 7
**Confidence:** 5

**Summary:**

This paper presents a multi-scale Vmamba model, which incorporates multi-scale information into the design of the Vmamba architecture. Additionally, the authors analyze how the attenuation coefficient between tokens increases as the modeled distance in Vmamba grows, whereas MSVmamba alleviates this attenuation issue by reducing the sequence length. The authors have conducted extensive experiments to thoroughly validate the effectiveness of MSVmamba. Moreover, the authors have also employed SE modules and ConvFFN to further enhance the model's performance.

**Strengths:**

1. The author's writing is clear and easy to follow.
2. The authors analyzed the issue of long-distance forgetting and proposed an effective method to address this problem.
3. The authors conducted comprehensive ablation experiments to validate the role of each module (MS2D, SE, ConvFFN) in MSVmamba for classification tasks.

**Weaknesses:**

1. Although MSVmamba has achieved better results compared to other Vmamba models, its scalability has not yet been validated. Could the authors provide training results of MSVmamba on larger models, such as those with 50M and 90M parameters, to demonstrate the model's scalability?

2. The authors' ablation experiments only validated the performance of the modules in classification tasks. It would be better to further verify the roles of each module in more fine-grained tasks such as detection and segmentation.

**Questions:**

Please refer to the weakness

---

> ### Author Rebuttal · Authors · 2024-08-06
>
> Thank you for your insightful comments and for the time you have dedicated to reviewing our manuscript. We greatly appreciate your feedback, which has been instrumental in enhancing the quality of our paper. Your concerns about the **scalability (Q1)** and the **ablation of our model in more fine-grained tasks(Q2)** are addressed as below.
>
> ### **Q1: Scalability.**
>
> Thanks for this valuable suggestion!  It’s truly important to validate the scalability of the proposed model. To compensate for this,  we have updated the results of the proposed model when scaling up to small and base size on ImageNet-1K. Please check further details in **Table 1** of the uploaded PDF. MSVMamba-S and MSVMamba-B consistently outperform the VMamba baseline by 0.6% and 0.6% top-1 acc respectively. Additionally, we are conducting further experiments on downstream tasks for small and base size models, including COCO detection and ADE20K segmentation. Due to time constraints, these results are still pending, but we commit to including them in the revised paper.
>
> ### **Q2: Ablation on more fine-grained tasks.**
>
> Thanks for this suggestion. As suggested, we have conducted an ablation study to evaluate the impact of each module within the detection and instance segmentation tasks on the COCO dataset, utilizing the Mask R-CNN framework. Detailed results of this study can be found in **Table 4** of the uploaded PDF. All experiments employed an ImageNet-1K pretrained backbone with a 100-epoch training schedule for initialization. The MS2D module alone brings improvements of 1.0% in box AP and 0.7% in mask AP compared to the VMamba baseline. Integrating additional components has further enhanced performance. The results on ADE20K segmentation are still undergoing due to the time limit. We will update all of the ablation details on fine-grained tasks in our revised paper.
>
>
> We hope that these revisions could address your concerns. **We thank you once again for your constructive feedback, which has significantly contributed to the improvement of our work.**

---

> > ### Comment · Reviewer_HsBE · 2024-08-11
> > **final reviewer**
> >
> > Thanks the author's efforts. My concerns are well addressed. I will keep my initial rates (7).

---

### Official Review · Reviewer_QVsi · 2024-07-13

**Soundness:** 2
**Presentation:** 2
**Contribution:** 1
**Rating:** 2
**Confidence:** 5

**Summary:**

This paper introduces a Multi-Scale Vision Mamba (MSVMamba) for computer vision tasks. It uses a multi-scale 2D scan operation on both the original and sub-sampled features to preserve long-range information and reduce computational costs. In addition, they address the problem of channel mixing in Mamba-based models by introducing a Convolutional Feed-Forward Network (ConvFFN) module. The resultant model achieves favorable performance on image classification and a variety of downstream tasks such as detection and segmentation.

**Strengths:**

1. The paper is well-written and easy to follow.

2. It attempts to address an important issue in making Mamba-based vision models more suitable for computer tasks. In order to achieve this, it introduces a hierarchical architecture, although not novel, but also integrates various modules such Convolutional Feed-Forward Network (ConvFFN) to further enhance the performance.

3. The analysis of effectiveness in 2D selective scan approach, initially introduced in VMamba, seems to be interesting and provide further insights on how to improve scan for vision tasks.

**Weaknesses:**

1. The major issue with this work is lack of novelty. Hierarchical Mamba-based vision models have been already introduced in VMamba and also its other variants. The MSVSS block seems to be a minor improvement over the existing VMamba block. In addition, the role of ConvFFN seems to be quite marginal. This is due to the fact that MLP blocks themselves can inherently perform channel mixing to a great extent.

2. Experiments are insufficient. This work only presents three small variants of MSVMamba-N, MSVMamba-M and MSVMamba-T with the biggest model having only 33M parameters. Hence, it is not really clear how the proposed approach scales for mid to larger sized models which have better accuracies. It even can't be compared to the small variants of many models (e.g. Swin-S) due to its contrived setting.

3. The paper only focuses on number of FLOPs as a representative for efficiency. However, a more practical scenario involved measuring throughput (or latency) on different devices (GPU, TPU, etc.). In particular, it is important to understand if the 2D selective scan approach introduces any significant overhead.


Post-rebuttal:

The following issues and weaknesses were revealed during rebuttal after interactions with the authors:

1. The proposed MSVMamba is slower than models such as ConvNeXt in both smaller and higher resolutions in terms of throughput (see Table 1 and Table 2 in rebuttal). This limits the practical usage of this work due to its lower throughput and can present significant challenges.

2. The authors deliberately presented results from the first version of VMamba which is not optimal. Although the second version (https://arxiv.org/abs/2401.10166v2) was released 42 days before the submission deadline, the authors claim that it should be considered as concurrent. This argument is not well-founded since we need to fairly evaluate the contribution of this work against VMamba and other methods.

3. The issue of limited novelty presents itself when comparing against VMamba (both first or second iterations). As expected, MSVMamba does not significantly improve the results -- authors claimed that MSVMamba addressed the long-distance forgetting issue in VMamba which is not backed up by these results.

Considering these issues, I lower my score to strong reject (2). I encourage the authors to revise their manuscript, include best results from VMamba and try to evaluate the contributions of their work quantitatively.

**Questions:**

1. How does the model compare in terms of image throughput to other Mamba-based vision models as well as CNN-based and ViT variants on a GPU ?

2. Is the 2D selective scan approach faster than the naive selective scan that is originally introduced in Mamba ?

3. Did authors try to scale up the model size to observe if performance is comparable to other models ?

**Limitations:**

Yes

---

> ### Author Rebuttal · Authors · 2024-08-06
>
> Thank you for your insightful comments and for the time you have dedicated to reviewing our manuscript. We appreciate your feedback, which has helped us improve the quality of our paper.  We address your concerns in a few points as described below:
>
> ### **Q1: Concerns About Novelty.**
>
> Thank you for this nice concern. We agree that the hierarchical architecture is not something new, as similar structures have been utilized in VMamba and various Vision Transformers. However, our contribution lies in the introduction of **an additional hierarchy within a single layer**, which we refer to as **"Hierarchy in Hierarchy"** in our title and is distinct from the hierarchical designs previously explored.
>
> Specifically, traditional hierarchical models that primarily focus on creating a feature pyramid between different stages. Our method makes the scanning process conducted on full-resolution and downsampled feature maps simultaneously, which introduces **a hierarchy inside one layer** or one stage. This **"Hierarchy in Hierarchy"** has not been explored in previous Vision Mamba-based works.
>
>  Besides, unlike **traditional multi-scale strategies** that primarily **enhance hierarchical feature learning**, our **MS2D** module is motivated by the need to **mitigate the long-range issue** prevalent in Mamba models. This novel approach is not merely structural but is specifically designed to tackle the long-range problem in selective scanning, a critical issue for Mamba-based vision models in computational tasks.
>
> The multi-scale 2D scan, though a straightforward strategy, effectively addresses this issue and achieves notable improvements. To further demonstrate its efficacy, we have conducted additional experiments on different scanning strategies and fine-grained tasks, the results of which are detailed in **Tables 3 and 4** of the uploaded PDF. Please check it for other details. Furthermore, ablations on tiny-size models with a 100-epoch training schedule are also reported **in table below**:
>
> | Model    | Param(M) | GFLOPs | Top-1 Acc(%) | Thru. (imgs/sec) | Train Thru. (imgs/sec) | Memory (MB) |
> |----------|----------|--------|--------------|------------------|------------------------|-------------|
> | VMamba   | 23       | 5.6    | 80.3         | 603              | 151                    | 6639        |
> | +MS2D    | 24       | 4.8    | 80.9         | 866              | 205                    | 4780        |
> | +Others  | 33       | 4.6    | 81.4         | 1092             | 331                    | 4532        |
>
> Our findings indicate that the proposed MS2D module contributes to an improvement of 0.6% in Top-1 accuracy for the tiny-size model. Other components of our model collectively contribute an additional 0.5% increase in accuracy. This ablation reveals that MS2D is more important in accuracy gain compared to other components on tiny-size model. Furthermore, the MS2D module not only enhances performance but also contributes to further speed gains and reductions in memory usage.
> In terms of the ConvFFN in our model, it is intended to maintain consistency with established methodologies, as detailed in lines 250 to 253 of our manuscript. We acknowledge that using an MLP is also a viable alternative. We apologize for any confusion this may have caused and will ensure to clarify this point more explicitly in the revised version.
>
> We hope this explanation helps to clarify the innovative aspects of our work and the specific challenges it addresses.
>
>
> ### **Q2: Insufficient experiments.**
>
> Thank you for this suggestion! It’s important to validate the scalability of the proposed model. To compensate for this, we have updated the results of the proposed model when scaling up to small and base size on ImageNet-1K. Please check further details in Table 1 of the uploaded PDF. Concretely, the MSVMamba-S and MSVMamba-B outperform VMamba by 0.6% and 0.6% top-1 acc. More experiments on downstream tasks are still undergoing due to the time limit. We will include more results of downstream tasks in the revised version.
>
>
>
> ### **Q3: Efficiency Comparison.**
>
> Thanks for this valuable suggestion! We apologize for the initial omission of a detailed efficiency comparison in our manuscript. To address this, we have complemented the efficiency comparison, including training/inference FPS and memory usage, with our baseline VMamba and widely-used SwinTransformer and ConvNeXt in **Table1 and 2** of the uploaded PDF for your reference.  Compared to our baseline, our proposed model achieves nearly **1.5x speedup in inference** and **2.0x speedup in training**. Additionally, it requires approximately **30% less memory**. The efficiency of our models at a 224x224 image resolution did not match that of well-established architectures such as Swin Transformer. However, when the image resolution is increased, our model achieves comparable efficiency to the Swin Transformer, which can be found in Table 2 of the uploaded PDF.
>
> In addition, we also complemented the efficiency comparison of tiny-size model between different scanning strategies in 2D selective scan and naive selective scan in Mamba with the same setting as Table1 of the uploaded PDF:
> | Model       | Param | Thru. (imgs/sec) | Train Thru. (imgs/sec) | Memory (MB) |
> |-------------|-------|------------------|------------------------|-------------|
> | Vallina Scan| 22.9  | 602              | 151                    | 6623       |
> | Cross Scan  | 22.9  | 603              | 151                    | 6639        |
> | MS2D        | 24.2  | 866              | 205                    | 4780        |
>
> It's worth noting that **no** FFN or ConvFFN is introduced in this comparison. As we can see, the cross scan in VMamba yields the same efficiency as the vallina scan in Mamba, while our multi-scan 2D scan further improves the efficiency .
>
>
> We hope these revisions could address your concerns and **thank you once again for your constructive feedback**!

---

> > ### Comment · Reviewer_QVsi · 2024-08-09
> > **Reviewer's Response to Rebuttal**
> >
> > I would like to thank the authors for providing responses to my feedback as well as uploading the rebuttal.
> >
> > I have the following concerns:
> >
> > 1. As mentioned in my other comment, the reported Top-1 accuracy from VMamba seem to be lower than their existing benchmarks. The authors claim that they have used the results from the first iteration of the VMamba arXiv submission. However, the benchmarks I refer to can also be found in the 2nd iteration which was available since Apr 10:
> >
> > https://arxiv.org/pdf/2401.10166v2
> >
> > Can the authors provide an updated version of Table 1 for VMamba and MSVMamba models alone (Tiny, Small and Base) with the updated Top-1 accuracy as reported in the above arXiv submission ?  it should suffice to just reply to my comment with this table.
> >
> > 2. The reported numbers in Table 2 for efficiency comparison are for batch size 32. Why not report numbers for standard 224x224 resolution with batch size 128 which matches the same setup in Table 1 ?
> >
> > Our goal here is to have a fair comparison to previous models to further understand the contributions of the proposed effort.

---

> > > ### Author Response · Authors · 2024-08-10
> > >
> > > Thanks for your feedback and we are very appreciated for this opportunity for further clarification. We will do our best to **achieve the goal of comparison to previous models to further understand the contributions of the proposed effort**.
> > >
> > >
> > > ### Q1: Why not include the results of the second version results in VMamba for comparison?
> > > We thank the reviewer for proposing this reasonable concern. As you mentioned, the second version of VMamba (VMambav9) was available since **Apr 10**, and the DDL of submission is **May 22**. According to NeurIPS 2024 official instruction, we argue that the second version of VMamba is not expected to be included in our comparison.
> > >
> > > The detailed evidence is provided by the NeurIPS 2024 official instruction (in NeurIPS-FAQ page). In Section **Submission format and content**, one question is “What is the policy on comparisons to recent work?”, and the given answer is “**Papers appearing less than two months before the submission deadline are generally considered concurrent to NeurIPS submissions.  Authors are not expected to compare to work that appeared only a month or two before the deadline.**” Thus, given the timeline of VMamba, we only include the first version of VMamba in our comparison.
> > >
> > > Besides, even if we take into consideration the contribution of VMambav9, our core contribution MS2D discusses and tackles a general long-range forgetting problem of VMamba, which is orthogonal to the contributions of other versions of VMamba. Thus, the appearance of existing VMamba variants does not weaken our contributions.
> > >
> > >
> > >
> > >
> > > ### Q2: Updated version of Table 1 for VMamba and MSVMamba models alone.
> > > As suggested, we add the updated version of Table 1 as below for your reference, which updated the results of the VMamba with **VMambav9** in the v2 version of VMamba paper.
> > >
> > > | Model       | Param(M) | GFLOPs | Top-1 Acc(%) | Thru. (imgs/sec) | Train Thru. (imgs/sec) | Memory (MB) |
> > > |-------------|----------|--------|--------------|------------------|------------------------|-------------|
> > > | VMambav9-T  | 31       | 4.9    | 82.5         | 1135             | 313                    | 5707        |
> > > | MSVMamba-T  | 33       | 4.6    | 82.8         | 1092             | 331                    | 4532        |
> > > | VMambav9-S  | 50       | 8.7    | 83.6         | 749              | 207                    | 5785        |
> > > | MSVMamba-S  | 51       | 9.2    | 84.1         | 665              | 232                    | 4910        |
> > > | VMambav9-B  | 89       | 15.4   | 83.9         | 542              | 151                    | 7918        |
> > > | MSVMamba-B  | 91       | 16.3   | 84.3         | 476              | 127                    | 6347        |
> > >
> > >
> > > As we can see, **VMambav9** exhibits **higher inference speed**  and **similar training speed** compared to our model.  However, its **Top-1 accuracy** on ImageNet still **lagged behind our model by 0.3%, 0.5% and 0.4%** for the tiny, small and base model respectively.
> > >
> > > ### Q3: Why Table 2 adopt a batch of 32 instead of a standard 128 for comparison?
> > >
> > > We apologize for this confusion as this explanation should be contained in the caption of Table 2. When testing the throughput, we follow the code provided in the official VMamba repo ( L580 in utils.py of analysis) and adjusted the batch size downwards to fit the GPU’s memory constraints. With the input resolution increase, the memory cost for models also increase significantly. Take Swin-tiny as an example, it takes more than 20000 MB memory with an input resolution of 768 and a batch size of 32. Adopting a batch size of 128 or 64 will cause Out-Of-Memory error in some model. To make all comparisons under the same configuration, we take a batch size of  32 in the Table 2 of the uploaded PDF as a compromise.
> > >
> > >
> > > We hope that these revisions could address your concern and we sincerely look forward to your feedback.

---

> > > ### Author Response · Authors · 2024-08-10
> > >
> > > Table: Comparsion of different versions of VMamba and our models.
> > >
> > > | Model       | Param(M) | GFLOPs | Top-1 Acc(%) | Thru. (imgs/sec) | Train Thru. (imgs/sec) | Memory (MB) |
> > > |-------------|----------|--------|--------------|------------------|------------------------|-------------|
> > > | VMamba-T [01-18]  | 23       | 5.6    | 82.2         | 603             | 151                    | 6639        |
> > > | VMambav9-T [04-10]  | 31       | 4.9    | 82.5         | 1135             | 313                    | 5707        |
> > > | MSVMamba-T  | 33       | 4.6    | 82.8         | 1092             | 331                    | 4532        |
> > > | VMamba-S [01-18]  | 44       | 11.2    | 83.5         | 425             | 106                    | 6882        |
> > > | VMambav9-S [04-10]  | 50       | 8.7    | 83.6         | 749              | 207                    | 5785        |
> > > | MSVMamba-S  | 51       | 9.2    | 84.1         | 665              | 232                    | 4910        |
> > > | VMamba-B [01-18]  | 76       | 18.0    | 83.7         | 314             | 77                    | 8853        |
> > > | VMambav9-B [04-10]  | 89       | 15.4   | 83.9         | 542              | 151                    | 7918        |
> > > | MSVMamba-B  | 91       | 16.3   | 84.3         | 476              | 127                    | 6347        |

---

> > > > ### Comment · Reviewer_QVsi · 2024-08-10
> > > > **Further Comment by Reviewer**
> > > >
> > > > I thank the authors for providing responses to my feedback.
> > > >
> > > > Regarding the throughput comparison, the authors mention:
> > > > " With the input resolution increase, the memory cost for models also increase significantly".
> > > >
> > > > However, my request was to use an input resolution of **224x224** and batch size of **128** which is very standard and has been presented in previous works before. This setup should be achievable on any consumer GPUs with moderate VRAMs.
> > > >
> > > > Can the authors only provide a comparison between MSVMamba, ConvNeXt and Swin Transformers in terms of throughput using this setup ? I would like to understand how your model compares to well-established models in the literature.

---

> > > > > ### Author Response · Authors · 2024-08-10
> > > > >
> > > > > Dear Reviewer QVsi,
> > > > >
> > > > > Thank you for your feedback. We apologize for any confusion regarding Table 2.
> > > > >
> > > > > To clarify, an input resolution of 224x224 and batch size of 128 has already been reported in Table 1 of the submitted PDF. If Table 2 were to report results for the same resolution and batch size, it would indeed be redundant with Table 1. Instead, Table 2 is intended to provide additional efficiency details across different resolutions.
> > > > >
> > > > > We hope this clarification addresses your concerns and helps to clear up any misunderstandings.

---

> ### Comment · Reviewer_QVsi · 2024-08-10
> **Comment by Reviewer**
>
> Thank you for your response. In this case, as shown in Table 1, there exists a substantial gap between the throughput of well-established models such as Swin and ConvNext and the proposed model.
>
> Even for larger resolutions, models such as ConvNeXt-T are significantly faster than the proposed MSVMamba-T. This may complicate the usage of this model where throughput is important (which is almost all use-cases at this time). And to add to this, ConvNeXt itself is not a very fast model in terms of throughput.
>
> Regarding the comparison between different versions of VMamba (released in less than two months after each other) and MSVMamba, the authors initially claimed that the 2nd iteration (available 42 days before submission deadline) should be disregarded. However, a later comment provided the performance for both models. Upon closer examination, we observe that MSVMamba does not significantly improve the Top-1 performance of the VMamba model.
>
> I appreciate the authors' efforts in this stage of the rebuttal. I believe I have all information I need to make a final decision.

---

> ### Comment · Reviewer_QVsi · 2024-08-10
> **Post-Rebuttal Score Update**
>
> I thank the authors for providing the rebuttal and their engagement during this period.
>
> Considering all aspects, I have decided to lower my score to strong reject (2). Here's the key reasons behind this decision:
>
> 1. The proposed MSVMamba is slower than models such as ConvNeXt in both smaller and higher resolutions in terms of throughput (see Table 1 and Table 2 in rebuttal). This limits the practical usage of this work due to its lower throughput and can present significant challenges.
>
> 2. The authors deliberately presented results from the first version of VMamba which is not optimal. Although the second version (https://arxiv.org/abs/2401.10166v2) was released 42 days before the submission deadline, the authors claim that it should be considered as concurrent. This argument is not well-founded since we need to fairly evaluate the contribution of this work against VMamba and other methods.
>
> 3. The issue of limited novelty presents itself when comparing against VMamba (both first or second iterations). As expected, MSVMamba does not significantly improve the results -- authors claimed that MSVMamba addressed the long-distance forgetting issue in VMamba which is not backed up by these results.
>
> I encourage the authors to revise their manuscript, include best results from VMamba and try to evaluate the contributions of their work quantitatively.

---

> > ### Author Response · Authors · 2024-08-11
> >
> > We thank Reviewer QVsi for the detailed summary in the post-rebuttal, where they pointed out three issues during the discussion. We would like to clarify each issue in details.
> >
> > Q1: The proposed MSVMamba is slower than models such as ConvNeXt.
> >
> > Note that our model design is specifically tailored for the Mamba architecture, allowing any subsequent optimizations related to efficiency to be directly inherited. Thus, we focused on comparing the VMamba baseline. It’s important to note that while our baseline VMamba is indeed much slower than ConvNeXt, it remains highly valuable and has inspired many subsequent works. For example, the first version of VMamba was released on 2024.01.18 and its citation is 300+ currently, which highlights its importance. Given the observation that our baseline VMamba is much slower than existing CNNs, we are motivated to improve its efficiency. Our proposed model achieves significant improvements, with nearly 1.5x speedup in inference and 2.0x speedup in training compared to our baseline VMamba. Taking other versions of VMamba into consideration, our contribution to efficiency is orthogonal to theirs, which could be integrated to achieve a better efficiency.  Concretely, our core contribution, the MS2D, focuses on the optimization of CrossScan in our baseline VMamba. In all three versions of VMamba, CrossScan inherently exists, and the speedup mainly comes from implementational optimizations and hyper-parameter adjustments in the Mamba block. Thus, our improvement is orthogonal to the subsequent techniques used in VMamba and can be integrated with them seamlessly.
> >
> >
> > Q2: The authors deliberately presented results from the first version of VMamba which is not optimal.
> >
> > As clarified in our previous comments, the second version of VMamba is released 42 days before the submission deadline and should be considered as concurrent work. This argument is well-founded by the NeurIPS 2024 official instructions (https://neurips.cc/Conferences/2024/PaperInformation/NeurIPS-FAQ). You can find the question “**What is the policy on comparisons to recent work?**” in the Section Submission format and content, and the corresponding answer is “**Papers appearing less than two months before the submission deadline are generally considered concurrent to NeurIPS submissions.  Authors are not expected to compare to work that appeared only a month or two before the deadline.**” Since 42 days is obviously less than 2 months, we do not include the second version of VMamba in our comparison and focus on the comparison with the first version.
> >
> > Although the second version of VMamba is defined as concurrent work based on the NeurIPS 2024 official instructions, we provide a detailed comparison with the second version as a reference during the rebuttal discussion because it is strongly recommended by Reviewer QVsi. Concretely, the Top-1 accuracy in the second version of VMamba on ImageNet still lagged behind our model by 0.3%, 0.5% and 0.4% for the tiny, small and base model respectively. This comparison highlights the superiority of our method over the concurrent work.
> >
> >
> > Q3: The issue of limited novelty presents itself when comparing against VMamba (both first or second iterations).
> >
> > In comparison with VMamba (the first version of VMamba), our models not only exhibit a 0.6% improvement in Top-1 accuracy on ImageNet across different model sizes but also show nearly 1.5x speedup in inference and 2.0x speedup in inference and training FPS.
> > The Multi-Scale 2D (MS2D) module, as the core contribution of our work, is well motivated by the need to mitigate the long-range issue prevalent in Mamba models. This is a critical challenge that has not been adequately addressed by existing methods. The subsequent optimization in VMamba focuses on the implementational optimizations and hyper-parameter adjustments in the Mamba block. Thus, the long-range issue inherently exists. VMamba utilizes multi-scan strategy with redundant FLOPs to alleviate this issue, while MS2D utilizes multi-scale strategy to efficiently tackle this issue. As our contribution in accuracy and efficiency gain is also orthogonal to the subsequent versions of VMamba, these techniques could be integrated for further improvement.
> >
> > We hope this clarifies our position and we appreciate your understanding.

---

### Official Review · Reviewer_XVcS · 2024-07-13

**Soundness:** 3
**Presentation:** 3
**Contribution:** 3
**Rating:** 7
**Confidence:** 4

**Summary:**

This paper presents a multi-scale vision mamba aimed at improving the performance of state space models (SSMs) in vision tasks while maintaining efficiency. The motivation stems from analyzing the multi-scan strategy in vision mamba, where the authors link its success to alleviating the long-range forgetting issue of SSMs in vision tasks. To address this problem effectively, the authors propose a multi-scale 2D scanning technique on both original and downsampled feature maps, which reduces the number of tokens in the multi-scan strategy. This method enhances long-range dependency learning and cuts down computational costs. Additionally, a ConvFFN is incorporated to overcome channel mixing limitations. Experimental results across various benchmarks validate the proposed multi-scale vision mamba's effectiveness.

**Strengths:**

The analysis of the multi-scan strategy's success in vision mamba is intriguing and could be valuable to the research community.
The proposed approach is straightforward yet effective, addressing the long-range forgetting problem while significantly reducing computational costs. The approach strikes a better balance between performance and FLOPs, as demonstrated in the experiments.
The paper includes comprehensive experiments on widely-used datasets and tasks, with comparisons to leading neural architectures showing the proposed networks' superiority.

**Weaknesses:**

The authors should consider including some simple baselines in additional ablation studies. For instance, besides the half-resolution branches in the proposed MSVMamba, reducing the scanning number in vision mamba would be a useful baseline. Incorporating these simple baselines could further highlight MSVMamba's effectiveness.
The paper outlines that the proposed MSVMamba uses 3 half-resolution branches and 1 full-resolution branch (Equations 8-11). However, the scanning direction for these branches is not clearly described. There is a lack of discussion on how the scanning direction for the full-resolution branch is chosen and how this choice impacts performance.

**Questions:**

Could you provide more comparisons with simple baselines?
Could you clarify the selection of different branches concerning the scanning direction and discuss its impact?

**Limitations:**

Reflecting on the weaknesses mentioned.

---

> ### Author Rebuttal · Authors · 2024-08-06
>
> Thank you for your insightful comments and for the time you have dedicated to reviewing our manuscript. We greatly appreciate your feedback, which helps enhance the quality of our paper. Below, we address your concerns regarding the **baseline involving the reduction of scanning numbers(Q1)** and the **scanning direction for the full-resolution branch(Q2)**.
>
> For clarity, we define the scanning directions as follows: Scan1 refers to the horizontal scan from the top-left corner to the bottom-right corner, and Scan2 refers to the vertical scan from the top-left corner to the bottom-right corner. Conversely, the reverse directions of Scan1 and Scan2 are denoted as Scan3 and Scan4, respectively.
>
> ### **Q1: Reducing the scanning number as additional baselines.**
>
> Thanks for the valuable suggestion. In response to this suggestion, we have added new baselines that involve only Scan1 (Uni-directional Scan) and a combination of Scan1 and Scan3 (Bi-directional Scan). These are now presented in **Table 3** of the uploaded PDF. Concretely, our MS2D further outperform **Uni-directional Scan** and **Bi-directional Scan** baselines by **3.0%** and **2.4%** top-1 acc respectively. We will include these results in the revised paper.
>
> ### **Q2: Ablation of scanning direction for full-resolution branch.**
>
> Thanks for your suggestion.  First of all, we apologize for the initial lack of clarity regarding the scanning direction in the full-resolution branch. In the original experiments, Scan1 was used as the full-resolution branch. To thoroughly explore the impact of different scanning directions, we conducted additional ablation studies as **table below**. The results indicate that while different scans yield similar accuracy, Scan1 was selected for its marginally superior performance consistency.
>
> | Full-res Scan | Scan1 | Scan2 | Scan3 | Scan4 |
> |---------------|-------|-------|-------|-------|
> | Top-1 Acc(%)  | 71.9  | 71.8  | 71.8  | 71.9  |
>
> These findings and the rationale for our choice of scanning direction will be updated in the revised paper to ensure clarity.
>
> We hope that these revisions adequately address your comments. **We thank you once again for your constructive feedback, which has significantly contributed to the improvement of our work**.

---

> > ### Comment · Reviewer_XVcS · 2024-08-12
> > **Thanks for addressing my concerns**
> >
> > I'd like to thank the authors' response to my question. I think the response is clear enough to address my confusion.

---

### Official Review · Reviewer_TQwz · 2024-07-13

**Soundness:** 2
**Presentation:** 2
**Contribution:** 1
**Rating:** 2
**Confidence:** 4

**Summary:**

The paper introduces a novel vision backbone model, MSVMamba, which incorporates State Space Models (SSMs) to address limitations in computational efficiency and long-range dependency capture in vision tasks. The model utilizes a multi-scale 2D scanning technique and a Convolutional Feed-Forward Network (ConvFFN) to improve performance with limited parameters.

**Strengths:**

The paper is easy to follow.

**Weaknesses:**

1. Lack of novelty.

The paper propose MSVMamba, the main contribution is shown in table.4. MS2D, SE, ConvFFN have already been proposed in previous papers. The main improvements come from existing knowledge.

2. Speed

The paper does not systematically measure the speed of their model on all the tasks and scales. The training and inference of MSVMamba could be slow compared with current models, like CAFormer, Conv2Former, CSwinTransformer.

3. Scalability

The model in the experiments is relatively small. Model with more than 300M or 600M parameters could show whether MSVMamba does perform better than current sota model. Now the experiments just show that the model converges quickly w.r.t. model size.

**Questions:**

Please refer to the weakness.

**Limitations:**

Please refer to the weakness.

---

> ### Author Rebuttal · Authors · 2024-08-06
>
> Thank you for your comments and the time you have dedicated to reviewing our manuscript. We appreciate your feedback, which helps us improve the quality of our paper.  We address your concerns in a few points as described below:
>
> ### **Q1: Lack of novelty.**
>
> We agree that multi-scale strategy, SE, and ConvFFN have been introduced in previous CNN or ViT-based works. However, the utilization of these techniques, especially the multi-scale strategy, is well-motivated by the analysis and empirical observation in this paper, which could provide insights to Mamba model design for vision tasks. Specifically, we take our core contribution MS2D as an example. To begin with, **MS2D is first introduced in this paper**, which is carefully designed for Mamba in vision tasks. Unlike traditional multi-scale strategies that primarily enhance hierarchical feature learning, our **MS2D module is motivated by the need to mitigate the long-range issue** prevalent in Mamba models. This is a critical challenge that has not been adequately addressed by existing methods. As detailed in lines 151 to 161 of our manuscript, our analysis demonstrates that the contribution of tokens significantly decays with increased scanning distance. To address this, we innovatively downsample the feature map to effectively shorten the sequence length. This adaptation not only introduces a multi-scale design within a single layer but also specifically targets the reduction of computational complexity and improves the efficiency of long-range interactions. Figure 4 and Table 4 in our paper provide compelling evidence of the effectiveness of our approach. The multi-scale design significantly alleviates the long-range problem, as visually demonstrated in Figure 4. Furthermore, the quantitative ablations presented in Table 4 underscore the substantial improvements our model achieves over existing techniques. Thus, **the introduced multi-scale method, although similar to previous works, originates from a different motivation and addresses a distinct problem**. We hope this explanation helps to clarify the innovative aspects of our work and the specific challenges it addresses.
>
> ### **Q2: Efficiency Comparison.**
>
> Thanks for the suggestion. We have complemented the efficiency comparison as suggested, including training/inference FPS and memory usage, with our baseline VMamba and widely-used SwinTransformer and ConvNeXt in **Table1 and 2** of the uploaded PDF  for your reference.
> We acknowledge that at the time of submission, the efficiency of our models at a 224x224 image resolution did not match that of well-established architectures such as Swin Transformer and ConvNeXt. However, it is important to highlight that the Mamba-based models still serve as a crucial backbone that achieves a competitive trade-off across various settings and is currently under continuous optimization. For instance, when the image resolution is increased, our model achieves comparable efficiency to the Swin Transformer, which can be found in Table 2 of the uploaded PDF. This is further supported by related experiments in ViM[1], which demonstrate the efficiency of the Mamba block in downstream tasks like detection.
> Our model design is specifically tailored for the Mamba architecture, allowing any subsequent optimizations related to efficiency to be directly inherited. In this work, our focus was primarily on comparing Mamba-based baselines. Our proposed model achieves significant improvements, with nearly **1.5x speedup in inference** and **2.0x speedup in training** compared to VMamba.
> We understand the importance of systematic speed measurements across all tasks and scales. While our current manuscript may not cover all possible configurations, we are committed to extending our evaluations and will consider including more comprehensive speed comparisons in future revisions or subsequent works.
>
>
>
>
> ### **Q3: Scalability**
>
> Thanks for the suggestion! We have updated the results of the proposed model when scaling up to small and base size on ImageNet-1K. Please check further details in Table 1 of the uploaded PDF. MSVMamba-S and MSVMamba-B consistently outperform the VMamba baseline by 0.6% and 0.6% top-1 acc respectively.
> When it comes to the model size that exceeds 300M, we acknowledge the potential benefits of testing larger models with parameters exceeding 300M to compare against current state-of-the-art (SOTA) models. However, it is important to note that most existing works involving vision mamba models [1,2,3,4,5] operate under 100M parameters. In this work, we strictly follow the setting of [1,2,3] to conduct evaluation and comparison. For example, the largest model in VMamba exhibits 76M parameters. Thanks for your valuable suggestion. We will include the evaluation of Mamba models with much larger sizes in future works.
>
> We hope these revisions could satisfactorily address your concerns and **thank you once again for your feedback**!
>
> **References:**
>
> [1] Zhu, Lianghui, et al. "Vision mamba: Efficient visual representation learning with bidirectional state space model." arXiv preprint arXiv:2401.09417 (2024).
>
> [2] Liu, Yue et al. “VMamba: Visual State Space Model.”  arXiv preprint arXiv:2401.10166v1 (2024).
>
> [3] Huang, Tao, et al. "Localmamba: Visual state space model with windowed selective scan." arXiv preprint arXiv:2403.09338 (2024).
>
> [4] Pei, Xiaohuan, Tao Huang, and Chang Xu. "Efficientvmamba: Atrous selective scan for light weight visual mamba." arXiv preprint arXiv:2403.09977 (2024).
>
> [5] Yang, Chenhongyi, et al. "Plainmamba: Improving non-hierarchical mamba in visual recognition." arXiv preprint arXiv:2403.17695 (2024).

---

### Official Review · Reviewer_kG7u · 2024-07-13

**Soundness:** 2
**Presentation:** 2
**Contribution:** 2
**Rating:** 5
**Confidence:** 4

**Summary:**

The paper introduces Multi-Scale VMamba (MSVMamba), a novel vision backbone model that leverages State Space Models (SSMs) to address the challenges of quadratic complexity in Vision Transformers (ViTs). The proposed Multi-Scale 2D Scanning (MS2D) and Convolutional Feed-Forward Network (ConvFFN) contributes to the final performance.

**Strengths:**

- This paper presents a multi-scale design of VMamba, reducing the sequence length by downsampling the input features.
- The SSM module proposed in this paper incorporated the strength of residual, dwconv, ssm, se-block.
- This paper achieves good results on classification and dense prediction benchmarks.

**Weaknesses:**

- This paper lacks the efficiency comparison among training/inference FPS and memory usage on real GPU devices.
- The proposed multi-resolution processing seems to contain too many inductive biases. I wonder if the gains would be reduced when applying to a standard-ablation-size model, i.e., VMamba-T.

**Questions:**

Please refer to the weakness

**Limitations:**

Yes

---

> ### Author Rebuttal · Authors · 2024-08-06
>
> Thank you for your insightful comments and for the time you have dedicated to reviewing our manuscript. We appreciate your feedback, which helps us improve the quality of our paper. Below, we address your concerns regarding the **efficiency comparison(Q1)** and the **ablation of our model to a tiny-size variant(Q2)**.
>
> ### **Q1: Efficiency Comparison Among Training/Inference FPS and Memory Usage.**
>
> Thanks for the suggestion. As suggested, we have complemented the efficiency comparison, including training/inference FPS and memory usage, with our baseline VMamba and widely-used SwinTransformer and ConvNeXt in **Table1** and **Table 2** of the uploaded PDF for your reference.  Compared to our baseline, our proposed model achieves nearly **1.5x speedup in inference** and **2.0x speedup in training**. Additionally, it requires approximately **30% less memory.**  The efficiency of our models at a 224x224 image resolution did not match that of well-established architectures such as Swin Transformer. However, when the image resolution is increased, our model achieves comparable efficiency to the Swin Transformer, which can be found in Table 2 of the uploaded PDF.
>
>
> ### **Q2: Ablation for tiny-size model.**
>
> Thanks for this nice concern. In response to this query about the performance of our model on a standard-ablation-size, we have conducted additional ablation studies with a 100-epoch training schedule. The results are detailed in the **table below** for your reference:
>
>
> | Model    | Param(M) | GFLOPs | Top-1 Acc(%) | Thru. (imgs/sec) | Train Thru. (imgs/sec) | Memory (MB) |
> |----------|----------|--------|--------------|------------------|------------------------|-------------|
> | VMamba   | 23       | 5.6    | 80.3         | 603              | 151                    | 6639        |
> | +MS2D    | 24       | 4.8    | 80.9         | 866              | 205                    | 4780        |
> | +Others  | 33       | 4.6    | 81.4         | 1092             | 331                    | 4532        |
>
>
> Our findings indicate that the proposed **Multi-Scale 2D (MS2D)** module contributes to an improvement of **0.6%** in Top-1 accuracy for the tiny-size model. **Other components** (SE, ConvFFN and N=1 in Table 4 of our paper) of our model collectively contribute an additional **0.5%** increase in accuracy. This ablation reveals that MS2D is more important in accuracy gain compared to other components on tiny-size model. Furthermore, the MS2D module not only enhances performance but also contributes to further speed gains and reductions in memory usage.
>
> We hope that these revisions could satisfactorily address your comments and **thank you once again for your constructive feedback**.

---

> > ### Comment · Reviewer_kG7u · 2024-08-13
> >
> > Thanks for your response. Your responses solve my concern to some extent. I maintain my score.

---

### Author Rebuttal · Authors · 2024-08-06

Dear reviewers and ACs:

First and foremost, we wish to express our sincere gratitude for the time and effort you have dedicated to reviewing our manuscript. Your insightful suggestion and comments could further enhance the quality of this paper.

We have conducted additional experiments to address the key points raised by the reviewers, as detailed in the uploaded PDF. Specifically:
- Table 1 addresses scalability concerns.
- Tables 1 and 2 complement the detailed efficiency comparison.
- Table 3 compares our approach with additional baselines.
- Table 4 explores ablations on fine-grained tasks.

Please refer to these tables for more detailed information. Thank you once again for your constructive feedback!

---

> ### Comment · Reviewer_QVsi · 2024-08-09
> **Discrepancies in VMamba Benchmarks**
>
> I'd like to thank the authors for providing the rebuttal. However, upon reading Table 1, VMamba Top-1 accuracy benchmarks seem to be  lower than the official release (and their paper):
> https://github.com/MzeroMiko/VMamba
>
> Any reason for this discrepancy ?

---

> ### Author Response · Authors · 2024-08-09
>
> Thank you for your reply. VMamba now has three versions on arxiv. The one you see is the third version released on May 26, which is latter than the submission ddl. The one compared in table 1 is the first version released on January 18, whose details could be found in 2401.10166v1. We hope this reply can clarify your doubts. And we sincerely look forward to your feedback.

---

> ### Comment · Reviewer_QVsi · 2024-08-09
> **Comment by Reviewer**
>
> I appreciate the authors' response, but why not using the best results/models when comparing to them ?
>
> The first version in Jan 18 was a preliminary effort which may not be optimized. And our goal is not to just show better performance, but also provide a meaningful comparison to previous efforts.
>
> The results I refer to can also be found in the second iteration of their arXiv paper which has been released since 10 Apr 2024:
>
> https://arxiv.org/pdf/2401.10166v2

---

> > ### Author Response · Authors · 2024-08-10
> >
> > We thank the reviewer for proposing this reasonable concern. As you mentioned, the second version of VMamba (VMambav9) was available since **Apr 10**, and the DDL of submission is **May 22**. According to NeurIPS 2024 official instruction, we argue that the second version of VMamba is not expected to be included in our comparison.
> >
> > The detailed evidence is provided by the NeurIPS 2024 official instruction (in NeurIPS-FAQ page). In Section **Submission format and content**, one question is “What is the policy on comparisons to recent work?”, and the given answer is “**Papers appearing less than two months before the submission deadline are generally considered concurrent to NeurIPS submissions.  Authors are not expected to compare to work that appeared only a month or two before the deadline.**” Thus, given the timeline of VMamba, we only include the first version of VMamba in our comparison.

---

> > > ### Comment · Reviewer_QVsi · 2024-08-10
> > > **Comment by Reviewer**
> > >
> > > I would like to thank the authors for providing their response.
> > >
> > > However, the argument that the "the second iteration of the VMamba paper (released 42 days before deadline) is concurrent and should not be considered while the first iteration is not concurrent" is indeed not well-founded and adds more confusion for other reviewers and readers.
> > >
> > > Even if we accept this work to Neurips, the authors MUST include the best results from VMamba. Hence, as a reviewer, it is our duty to evaluate the effectiveness of this work in a fair manner which includes reasonable comparisons to previous efforts.

---

> > > > ### Author Response · Authors · 2024-08-10
> > > >
> > > > Dear Reviewer QVsi,
> > > >
> > > > We sincerely appreciate your prompt feedback and the opportunity to clarify our position. We apologize for any confusion regarding the evidence we presented earlier. For further clarification, we update the concrete link in our previous comment here. The official NeurIPS 2024 FAQ (https://neurips.cc/Conferences/2024/PaperInformation/NeurIPS-FAQ) states:
> > > >
> > > > "**Papers appearing less than two months before the submission deadline are generally considered concurrent to NeurIPS submissions**"
> > > >
> > > > This definition of concurrent work is provided by the official NeurIPS instructions, not by us. We adhered to these guidelines in our comparisons and citations.
> > > >
> > > > We are genuinely grateful for your strong sense of responsibility in reviewing our work. We hope this explanation resolves any misunderstandings.
> > > >
> > > > Thank you again for your thorough review and commitment to maintaining high standards in academic research.
> > > >
> > > > Sincerely,
> > > > Authors of Paper 3968

---

> > > > ### Author Response · Authors · 2024-08-10
> > > >
> > > > Dear Reviewer QVsi,
> > > >
> > > > We respectfully agree with your opinion that we must include the best results from VMamba.
> > > >
> > > > To address this concern and provide further clarification, we have uploaded a detailed comparison of different versions of VMamba and our models in the rebuttal comment box for your reference. This table offers a comprehensive overview of the performance metrics across various model versions.
> > > >
> > > > We hope this additional information helps to clear any misunderstandings. Furthermore, we commit to including the best results from VMamba in our revised paper to ensure a thorough and fair comparison.
> > > >
> > > > We appreciate your diligence in reviewing our work and look forward to your feedback on this updated information.

---

> ### Comment · Reviewer_XVcS · 2024-08-12
> **Please allow me to step forward and share my thoughts on this discussion thread**
>
> VMamba is a new and promising backbone network in computer vision that warrants more research efforts. However, from what I understand, VMamba-based models currently face challenges in achieving better efficiency than CNNs and ViTs. This is why numerous recent works aim to further enhance Mamba to make it viable for real-world applications. In my opinion, the authors’ claim of achieving a 1.5x speedup in inference and a 2.0x speedup in training is commendable and meets expectations.
>
> I reviewed the NeurIPS 2024 FAQ for Authors on the official website, and it appears that the second version of VMamba, which appeared on arXiv this April, can indeed be considered concurrent with this submission. Therefore, I have no concerns about the current comparative experiments, and it’s great to see the authors include new comparison results with that second version of VMamba.
>
> Overall, I am inclined to accept the paper and am open to further discussions with other reviewers.

---

### Decision · Program_Chairs · 2024-09-25

**Decision:**

Accept (poster)

**Comment:**

The paper proposes Multi-Scale VMamba, a hierarchical visual state space model that aims to address the limitations of existing models by improving efficiency and long-range dependency learning through a multi-scale 2D scanning technique and a Convolutional Feed-Forward Network (ConvFFN). The reviewers generally appreciated the problem’s importance and the proposed solutions but raised concerns regarding the novelty of the approach, given the concurrent work on VMamba, as well as scalability and efficiency in practical applications and more detailed ablations.

In their rebuttal, the authors provided new results on scalability with small and base versions, further results on efficiency, additional ablations to clarify the usefulness of proposed components. They also argued that VMamba-v2 should be considered concurrent work, citing NeurIPS guidelines that suggest work appearing less than two months before the submission deadline should not be mandatory for comparison. The authors made a case that their contributions are orthogonal to the improvements in VMamba to which most of the reviewers agree.

After carefully considering the reviews, the authors’ rebuttal, and the discussion, the AC acknowledges that the proposed approach has merit and presented ideas will be useful to the efforts to improve efficiency and performance of Mamba architectures. Specifically, not including VMamba-v2 in comparisons is understandable given the work is continuously improving and the specific changes were close to the submission dates (however, a more direct comparison with SoTA implementations will improve this work's quality - kindly see the comments below). The AC also recognizes the efforts made by the authors to address the reviewers’ concerns, particularly with the additional experiments and comparisons. While some concerns remain, particularly regarding scalability and efficiency, the paper presents a meaningful contribution to the field by tackling the long-range dependency issue in state space models with an interesting approach.

Considering all aspects, the AC recommends an accept decision. The authors are encouraged to further clarify their contributions and comparisons with VMamba in the final version and include all suggested changes. In line with the author's comment, "our contribution to efficiency is orthogonal to theirs, which could be integrated to achieve a better efficiency," it will be a great addition to the final version if the authors can include their proposed MS2D and ConvFFN on top of the recent version of VMamba to demonstrate its generalization and ability to provide gains over more competitive baselines. Similarly, AC will strongly suggest authors to update Fig. 1 comparison with actual GPU runtimes on x-axis, since the FLOPs comparison can give a misleading picture.